# Strengthening Mechanism of Polyurea to Anti-Penetration Performance of Spherical Cell Porous Aluminum

**DOI:** 10.3390/polym16091249

**Published:** 2024-04-30

**Authors:** Zhiqiang Fan, Yujian Guo, Yongxin Cui, Xiaopeng Yang

**Affiliations:** 1School of Aeronautics and Astronautics, North University of China, Taiyuan 030051, China; guoyujian1017@163.com (Y.G.); cuiyongxin0128@163.com (Y.C.); yagnxiaopeng2024@163.com (X.Y.); 2State Key Laboratory of Dynamic Measurement Technology, North University of China, Taiyuan 030051, China; 3School of Mechanical and Electrical Engineering, North University of China, Taiyuan 030051, China

**Keywords:** anti-penetration, fragment, polyurea coating, spherical cell porous aluminum

## Abstract

A composite structure containing a metallic skeleton and polyurea elastomer interpenetrating phase was fabricated, and its anti-penetration performance for low-velocity large mass fragments was experimentally studied. The protection capacity of three polyurea was compared based on the penetration resistance force measurement. Results show that the polyurea coating layer at the backside improves the performance of the polyurea-filled spherical cell porous aluminum (SCPA) plate due to its backside support effect and phase transition effect, which are accompanied by a large amount of energy absorption. The frontal-side-coated polyurea layer failed to shear and provided a very limited strengthening effect on the penetration resistance of the interpenetrating phase composite panel. The filling polyurea in SCPA increased the damage area and formed a compression cone for the backside coating layer, leading to a significant stress diffusion effect. The anti-penetration performance was synergistically improved by the plug block effect of the interpenetrating phase composite and the backside support effect of the PU coating layer. Compared with SCPA, the initial impact failure strength and the average resistance force of the composite plate were improved by 120–200% and 108–274%, respectively.

## 1. Introduction

Considering the extreme loading conditions, such as blast fields, the protective structures are usually under the coupling action of shock waves and high-speed fragments, which has a nonlinear superposition damage effect on the structure [1]. Therefore, materials and structures possessing high resistance to both shock waves and fragments have attracted great interest in the past decades [2,3,4]. Cellular materials such as foams, honeycombs, and lattices have been widely used as core layers in light-weight composite structures because of their high porosity and energy dissipation performance [5,6,7]. These porous materials also have high shielding performance against heat, electromagnetic waves, and sound waves. But its anti-penetration capacity for fragments and projectiles is relatively weak due to its porous structure and low density. Metallic [8], ceramic [9,10], and fiber-reinforced composite materials [11] with high strength and toughness are more widely used for impact protection from projectiles and fragments. However, studies on composite structures containing both energy absorption layers and anti-penetration layers are still lacking due to the complex matching mechanisms of strength, stiffness, energy absorption, and anti-penetration performance between different layers. In addition, metallic and ceramic layers would lead to a significant increase in weight, which would even adversely affect the function of the composite structure.

Polyurea coating is a newly developed protective technology for structure design that has exhibited excellent blast and impact mitigation performance. Polyurea has a very obvious nonlinear stress–strain response and strong strain rate sensitivity. It also has an adjustable elastic modulus, tensile strength, and ductility [12]. The strong adhesion to different material interfaces has significantly promoted its application in the protective performance enhancement of concrete slabs [13,14], metal plates [15,16,17], ceramic panels [18,19], porous materials [20,21] and fiber-reinforced composite structures [22,23,24] to blast and impact. These studies have indicated that polyurea layers can not only restrain metal plate deformation with their high dynamic strength and toughness but also increase the energy absorption capacity through glass transition and large ductile deformation [25,26,27].

In addition to the use of anti-explosion performance enhancement, the application of polyurea in anti-penetration protection is more extensive. Numerous studies on polyurea-coated aluminum and steel plates have revealed that polyurea layers contributed positively towards the reduction in residual velocity and increase in ballistic limits [28,29,30]. The strengthening effects of polyurea-coated metal panels were concluded to be glass transition, self-closing [31], cracking, spallation, and local fragmentation mechanisms, and the backside coating method towards a better enhancing effect [25]. The polyurea coating on ceramic panels can also improve ballistic protection by reducing energy consumption during the glass transition of polyurea [15], which can prevent local failure of the ceramic tiles and increase the perforation damage volume of the ceramic cone. But the advantage shown is a non-monotonous relationship with thickness, and the front side coating is proposed as the best configuration, while the sandwiched polyurea layer even weakens the protective performance of composite armor [17].

Compared to high-speed penetration, studies on the anti-impact resistance of polyurea-coated panels to large mass and low-velocity projectiles are relatively lacking. Ao [22] conducted quasi-static and low-velocity (4.5 m/s) indentation tests on polyurea-coated CFRP laminates. The specific energy absorption under dynamic and static indentation increased by 94% and 51%, respectively, which are largely caused by the increased damage area and fracture crack. The front surface coating was the best method. In addition, it was demonstrated that the bonding strength does not affect the peak force and fracture displacement but only affects the force drop and fracture [32]. Jiang [33] studied the impact response of polyurea-coated steel plates to a low-velocity cylindrical hammer. The frontal side coating showed the best energy absorption and anti-indentation performance. Wang [34] experimentally studied the impact response of a polyurea-coated ceramic–aluminum composite plate to low-velocity (100–300 m/s) and large mass (100 g) fragments. Unlike high-velocity projectile impacts, low-velocity and large-mass fragments will flip in varying degrees after hitting the target.

As seen, the polyurea coating layer can not only produce phase transitions in its own molecules and absorb a large amount of impact energy, but it can also change the failure mode of matrix plates, producing more failure patterns and a larger damage area. But the strengthening mechanisms for different materials vary greatly. Apart from the divergence of the coating position, the effect of the coating thickness also shows a non-monotonous effect on ballistic protection [3,16,35], indicating that the protection mechanism is very complex and unclear.

For energy absorption material, such as aluminum foams, the strengthening mechanism of polyurea coating may be even more complex due to the special material structure and mechanical behavior of porous materials. The porous surface topography of aluminum foam could largely enhance the interface bonding strength. When the structure is penetrated by projectiles and fragments, the metallic skeleton could also help to change the moving route, thus improving the anti-penetration performance. Research work investigating and describing the behavior of polyurea-coated porous materials under ballistic impacts tends to be limited. Bijanzad [18] investigated the enhancement effect of polyurea coating on the ductility, plastic deformation, and fracture toughness of aluminum foams with different densities. The bending tests demonstrated a significant increase in both failure load and strain. Also, spraying polyurea coating on the 3D auxetic lattice can increase the elastic ultimate stress of the rods due to the strong wrapping effect. The coating layer distributed stress more evenly throughout the core layer, thus leading to higher energy absorption performance [19].

Apart from the spray coating method, porous materials can benefit a lot from filling with polymer foams by improving energy absorption and shock resistance [36,37,38]. Metallic skeletons filled with functional polymers usually have two constituent phases that are interconnected throughout the micro-structure. The different constituent phases in interpenetrating phase composites (IPCs) [39] could contribute their properties to the overall macro-scale characteristics synergistically, and these coupling effects result in more advantages. Liu [40] prepared IPCs by filling the spherical cell porous aluminum (SCPA) with polyurethane to enhance the damping and hysteretic friction capacity of metallic foams. The IPCs have a good potential as friction dampers, and the polymer filler with hyper-elastic deformation could compensate for the disadvantage of SCPAs that are not affected by the recoverable deformation in the stress plateau stage [41,42]. Fan [43] prepared an IPC by involving rigid polyurethane foam in SCPAs and studying the dynamic energy absorption mechanism. It was demonstrated that the PUR filler significantly increased the specific energy absorption without decreasing the compressibility, and it also improved the strain rate dependence of the plateau stress. Even though the energy absorption and anti-fatigue performance of IPCs containing metallic skeletons and polymer fillers have been studied, the anti-penetration characteristics are still unclear when considering the combined impact of shock waves and fragments. The filling and coating of polyurea elastomer for porous aluminum panels provide a new approach to improving the anti-penetration performance of energy absorption layers in protective structures. The composite plate containing both metallic skeleton and polyurea coatings could provide both anti-blast and anti-penetration capacities, showing high potential in armor design for personal protective equipment and light armored vehicles.

In consequence, this study aims to identify the penetration response of a novel metallic skeletal IPC with polyurea filling and coatings under the low-velocity impact of large mass fragments. Experiments were conducted on polyurea-filled SCPA panels coated with polyurea layers. The effect of polyurea types and impact configurations on anti-penetration performance was studied. The anti-penetration resistance was measured, and the enhancing mechanism of polyurea coating on SCAP composite panels was analyzed. The experimental results could help to develop advanced anti-strike panels possessing high protection performance from blast loading and fragment penetration, which is of high potential in the structural design of personal protective equipment and lightweight armored vehicles.

## 2. Materials and Methods

### 2.1. Materials

The spherical cell porous aluminum (SCPA) specimens were prepared through the space holder approach and provided by Qiangye Metal Foam Ltd., Taiyuan, China, as illustrated in [44,45]. The matrix material is pure aluminum, Al 99.7%. The average diameter of a spherical cell is about 6 mm, and each spherical cell has 4~6 openings with a size of 1~2 mm in different directions, as shown in Figure 1a. The SCPA was composed of a uniform inter-connective open-cell porous structure system. The side length and thickness of the SCPA plate are 200 mm and 30 mm, respectively. The space holders are of a unified diameter and closely stack inside the mold before the pouring of the melted aluminum liquid. The theoretical porosity of the particle accumulation is about 30–39%, while the porosity of SCPA is about 61–70%. The apparent density of the SCPA plate used in this work was measured before the filling and coating process, the density was about 0.85–0.95 g/cm^3^. The actual porosity of SCPA is 64.8–68.5%, which is in accordance with the theoretical results.

The polyurea elastomer was provided by Jindun Protection Technology Co., Ltd., Jinan, China. Three elastomers were filled into different SCPA plates, and the mechanical properties of polyurea are illustrated in Table 1. According to the previous studies [46,47], the single-sided coating of polyurea was better than the double-sided coating for deformation mitigation. Therefore, all testing specimens were unilaterally sprayed with APC-40 polyurea, and the thickness of the coating layer was 3.0 mm.

The SCPA/polyurea elastomer interpenetrating phase composite (IPC) structure was fabricated through both infiltration and spray methods. Figure 2a illustrates the fabrication process of PSCPA panels, including filling, spraying, and cutting processes. Firstly, the SCPA plate was cleaned and placed into the mold made of teflon panels. Then the PU mixture was poured into the mold and pushed into the spherical cells by the piston. It should be noted that the SCPA plates are placed vertically to facilitate flat spraying surfaces on the front and backside surfaces. In this work, three polyurea elastomers (labeled APC-20, APC-30, and APC-40) were filled into different SCPA plates to compare their anti-penetration performance. Secondly, the PU-filled SPCA place was taken out of the mold before the polyurea filled into SCPA cells fully consolidated, and then the polyurea coating APC-40 was sprayed onto the newly formed surface, as shown in Figure 2a. The coating layer was formed by multiple spraying processes until the thickness reached 3 mm. The composite plate containing the SCPA skeleton, elastomer filler, and polyurea coating layer is shown in Figure 1b. Composite plates filled with three polyurea were labeled PSCPA-I, PSCPA-II, and PSCPA-III in the following analysis. The typical sectional view of the PSCPA panel is shown in Figure 2b–d. As seen, the interfaces between the aluminum skeleton, filled phase, and the coating layer show a good bonding state. In addition, there were large amounts of micro-pores in the coating layer due to the spraying formation technique.

### 2.2. Methods

The penetration experiment was conducted using a 50 mm diameter one-stage gas gun, as illustrated in Figure 3a. Two groups of tests were implemented: The first group was designed to measure the penetration resistance at low impact velocity, and the other group aimed to study the anti-penetration mechanism of the composite structure for large fragments at relatively high impact velocity. In the first set of tests, the drop-weight measurement technique was applied; that is, the penetration-resistant force was acquired by measuring the acceleration of the mass block. However, the difference from the traditional drop-weight test is that the mass block was accelerated by high-pressure gas to obtain a higher impact velocity than a free-falling weight. As seen in Figure 3a, the projectile was launched by the gas gun to acquire the initial penetration velocity.

The sample was fully clamped onto the bracket, which has an 85 mm × 85 mm deformation window. The projectile system consists of a mass block and an incident bar, as shown in Figure 4a. Two accelerometers were fixed on the tail part of the mass block, and the acceleration of the projectile system was calculated by averaging the measurement results of two sensors. The diameter and length of the mass block were 50 mm and 150 mm, respectively. There was a 3 mm wide groove along the axial direction of the mass block to protect the wires of the accelerometer, as seen in Figure 4b. The diameter and length of the incident bar were 14.5 mm and 250 mm, respectively. Moreover, two bars with different head shapes, that is, a spherical nose (SN) and a flat nose (FN), were applied to study the effect of fragment shape on penetration resistance, as shown in Figure 3b. The total mass of the projectile system is about 2.8 kg, and the initial impact velocity was measured at 15–17 m/s by the laser velocimeter at the front end of the gun barrel.

To avoid the mass block directly impacting the sample, a large cylindrical mass stopper was placed on the backside of the sample to absorb the residual kinetic energy of the projectile system. The stopper was a steel bar with a dimension of ∅50 mm × 800 mm, which was constrained by two lead rails. Once the incident bar penetrates the backside of the sample, the projectile system is decelerated by the impact between the incident bar and the stopper, and the residual energy is converted to the low-velocity motion of the large mass stopper. In this way, the axial motion of the mass block was limited to the gun barrel to protect the measuring system.

Considering the penetration behavior of the incident bar was significantly affected by the mass block due to the radial constraints of the gun barrel, cylindrical fragment penetration tests were also conducted. The dimension of the fragment was ∅14.5 mm × 55 mm, and the weight was about 70 g, as shown in Figure 4c. The fragment was installed into a plastic sabot with dimensions of ∅49.5 mm × 60 mm. The sabot was made of polylactic acid (PLA) by using 3D printing technique, as seen in Figure 4d. The sabot was detached by high-velocity impact onto a rigid separator in front of the sample. In high-velocity penetration tests, the initial impact velocity of the fragment was about 155–170 m/s. Two repeated experiments were performed for each testing condition to acquire the average resistant force and accurately evaluate the anti-penetration capacity of different panels.

## 3. Results

### 3.1. Penetration Resistance

The typical testing results of the first group tests are shown in Figure 5, including the penetration result of the incident bar and the force–displacement curves, which represent the impact load-carrying capacity of the specimens. As seen in Figure 5a, the incident bar passed through the SCPA panel and was further stopped by the energy absorber. While for the PSCPA-II sample, the incident bar was stuck in the panel, indicating that the composite panel exhibited much higher resistance than SCPA. It should be noted that the PU coating layer was placed on the backside of the sample to better utilize its limiting effect on bending deformation.

Figure 5b,c shows the resistance force curves of SCPA and PSCPA-III subjected to flat nose bar impact, respectively. As seen, the resistance curve can be divided into three parts corresponding to different penetration stages: The transient stage, plateau stage, and unloading stage. The first transient stage corresponds to the impact between the incident bar and the front surface of the panel, showing a quickly increased force and relatively small displacement. In the plateau stage, the incident bar progressively destroys the inner cells and even penetrates the panel, causing slowly increasing penetration resistance. For SCPA, the resistant force increased from 6.5 kN to 8.3 kN during the penetration. The high-frequency fluctuation on the curve was mainly caused by the resonance oscillation of the accelerometer and the intermittent impact between the incident bar and the layered cells. As the penetration depth exceeded 20 mm, the force quickly decreased with the displacement because of the tearing fracture of the backside cells of SCPA, as shown in Figure 5b. Therefore, the third stage corresponds to the breakthrough process of the incident bar from the SCPA panel. The red line presents the velocity-decreasing process. As seen, the residual velocity was about 10.8 m/s after the incident bar passed through the panel. The energy dissipation of the SCPA panel was about 151.7 J during the penetration of the projectile system. The backside view of the penetration hole is shown in Figure 6, and the outlet diameter was about 25 mm. It was demonstrated that the incident bar also broke through the PU coating layer of the PSCPA-I panel, while the coating layer stopped the penetration of the incident bar for the PSCPA-II and PSCPA-III panels.

Figure 5c,d indicates the resistance force curves of the PSCPA-III plate under the impact of the flat and spherical nose incident bars, respectively. The force curve can also be divided into three stages, but the response shows some different characteristics compared with that of SCPA. As seen in Figure 5c, the strength failure force in the first stage is about 17.5 kN, which is much higher than that of SCPA. This should be attributed to the strengthening effect of PU filler in the spherical cells. In the second stage, the penetration resistance increased with the displacement due to the incompressibility of cells in the composite panel, and the average resistance of PSCPA-III increased by about 202.7% compared with that of SCPA. In the third stage, the force decreased quickly with the displacement, corresponding to the penetration termination stage. In this stage, the impact energy is gradually dissipated by the destructive cells and the deformed PU coating layers. The final penetration depth was only about 13.5 mm, and the PU coating produced a 4.6 mm high hump, as seen in Figure 6.

For PSCPA-III impacted by the spherical nose (SN) incident bar, the initial failure strength is about 12 kN, and the average penetration resistance is 17.5 kN, as shown in Figure 5d. The final penetration depth is 20 mm, and the high impact caused by the PU coating produced an 11 mm high hump, as seen in Figure 6d. The high tensile strength and elongation of the PU coating under dynamic loading significantly decreased the bending deformation of the SCPA plate and prevented the penetration failure. The energy dissipation of PSCPA-III under flat and spherical nose bar impacts was 240 J and 272 J, respectively.

Figure 7 shows the comparison of penetration resistance obtained from SCPA and PSCPA-III subjected to flat nose and spherical nose incident bars. As seen, the resistance force under the flat nose bar penetration was much higher than that of the spherical nose bar due to the larger impact interface during the penetration process. The average resistance of SCPA under two different bars was 7.14 kN and 5.48 kN, respectively, while for panels filled with PU-III, the average resistance increased to 22.68 kN and 17.5 kN. Figure 7c also shows the repeated testing results for PSCPA panels. The curves show good repeatability for the same test condition, indicating that the measurement and testing methods for evaluating the anti-penetration capacity of composite panels have high reliability.

The effect of the PU type on the penetration resistance of the composite plate is displayed in Figure 8, and the average resistance force is summarized in Table 2. As seen, the initial failure strength of the composite panel is about 2.2–3.0 times that of SCPA; the average resistance is improved by 108–218% and 125–274% for FN and SN penetration conditions, respectively. The force–displacement curve of SN penetration shows a longer, increasing process due to the gradually increased contact surface between the incident bar and the panels. The increase in penetration force at the plateau stage is largely attributed to the complex strengthening mechanism of the PU filler on the metallic skeleton.

### 3.2. Fragment Penetration

From the comparison of the penetration resistance of three composite panels, the panel filled with PU-III has the highest resistance force to fragment penetration. Therefore, high-velocity impact tests of the cylindrical fragment were conducted on the PSCPA-III sample, and the SCPA plate was also tested for comparison. According to previous studies, it was concluded that the PU coating layer exhibits better anti-penetration performance when sprayed on the frontal side surface of the substrate panels [48], while it exhibits a better deformation mitigation effect at the backside when the substrate is subjected to large-scale blast impact [4]. Therefore, the impact direction of the composite plate was taken into consideration; that is, impact tests with the PU coating layer on the front side and the backside were both conducted. The cylindrical fragment was fixed into the plastic sabot and launched by the one-stage gas gun; the impact velocity was about 155–170 m/s. A typical testing result for SCPA with an incident velocity of 155 m/s is shown in Figure 9a. As seen, the SCPA panel was severely destroyed by the fragment. The fragment passed through the panel with a residual velocity of about 115 m/s, which was measured by a high-speed camera. In addition, the fly direction of the cylindrical fragment after passing through the panel was significantly influenced by the porous structure of the SCPA. A series of discontinuous impacts between the fragment and the cell walls changed the penetration path in the plate and further improved energy dissipation performance.

Figure 9b shows the reverse impact result of PSCPA-III; that is, the impact face was the PU coating layer while the PU-filled porous aluminum was at the backside of the panel. The fragment penetrated the composite panel with a relatively low residual velocity of about 12 m/s. The PU coating layer was sheared by the fragment, producing a small entrance due to the self-closing effect, which formed the intricate overlap of tensile deformation filaments in the perforation [49]. However, for the PU-filled porous aluminum plate, the fragment caused large bending deformation and tearing failure. Spherical cells and PU fillers in the penetration channel were destroyed and rushed out by the fragment. Several radial cracks were observed around the destruction zone, which were attributed to the low tensile strength of SCPA.

As a comparison, Figure 9c shows the forward impact result of PSCPA-III with an incident velocity of 161 m/s. It was found that the fragment was embedded into the panel, and the coating layer produced a cone hump at the back surface. The special micro-structure of PSCPA affects the penetration direction of fragments, and the high-strength PU coating restrains the bending deformation of porous plates, leading to higher anti-penetration performance than reverse impact.

## 4. Discussion

To analyze the anti-penetration mechanism of SCPA and PSCPA panels, the tested specimens were sliced, and the sectional images of the specimens were acquired. Figure 10a shows the sectional view of the penetration channel of the SCPA plate, while Figure 11 and Figure 12 correspond to PSCPA-III panels subjected to reverse and positive impacts, respectively. As seen in Figure 10a, the spherical cells mainly failed because of the shear force caused by fragment penetration. The porous structure resulted in a low resistance to shear strength in the material, and the cells in the channel were destroyed by shear before compression. The diameters of the entrance and exit of the penetration channel were 15 mm and 29 mm, respectively. The expansion of the channel along the penetration direction was largely attributed to the plug-blocking effect of the compressed cells. But the plug effect is very limited because of the compressibility of cells and the unconstrained back surface of the SPCA plate. In consequence, the fragment easily penetrated the panel with low energy dissipation. Figure 10b shows the SEM image of the fracture surface of the cell wall in the penetration channel. As seen, the fracture of the cell walls mainly shows a tearing failure pattern, and the slipping imprints also indicate the ductile failure of the aluminum skeleton.

Figure 11a shows the reverse impact results of PSCPA-III plates; the failure pattern under low-velocity impact was also displayed for comparison, as seen in Figure 11b. The PU coating on the front surface was peeled off the PSCPA plate during the cutting process. As seen, the high-velocity fragment mainly caused shear failure on the PSCPA plate; the exit diameter is larger than the incident hole. The spherical cells near the penetration channel at the front part of the PSCPA plate also produced a compression–shear complex deformation pattern, as the yellow arrows indicate in Figure 11a. However, under low-velocity impact of the fragment, the PU coating layer firstly resisted the penetration by producing large tensile deformation, and then the impact energy was further dissipated through the bending deformation of the PSCPA panel. The spherical cells and PU fillers at the backside part of the plate failed in tearing mode, and the damage zone spread to a larger range due to the global bending deformation of the panel. The cells at the incident path were largely compressed by the fragment, while they failed by shearing around the channel. The PU fillers have shown a limited strengthening effect on the compression and tear properties of the spherical cells due to the limited connections through the openings around the spherical cells, but they can increase the lateral expansion effect of the damaged part because of their uncompressibility. But once the penetrable crack formed through the metallic skeleton, large pieces of PSCPA fragments were rushed out, and the structure was finally damaged by bending failure. Figure 11c shows the micro-scanning image of the fracture surface of the PU coating layer. As shown, the coating layer contains a large amount of micro-pores, which lead to perforative fracture under tensile stress.

The sectional view of the forward impact result of PSCPA-III is shown in Figure 12a. More failure patterns can be observed, including the fan-shaped damage zone, the local deformation and peeling failure of the PU coating layer, and the severely crushed plug block. The spherical cells adjacent to the incident hole were mainly crushed by compression and shear stress, while cells near the diffusion boundary of the fan-shaped damage zone were more destroyed by tensile and shear stress. The sheared fragments of spherical cells and PU fillers were severely crushed and formed the plugged block, which significantly increased the contact area between the PSCPA panel and the coating layer. The increased contact surface largely dispersed the local stress imparted on the panel and further converted the impact energy into stretching deformation energy, acting like the ceramic cone effect [50]. The stress dispersion mechanism largely restrained the penetration of the PU coating layer and improved the anti-penetration performance of the composite panel. The stretched PU coating layer acts as a buffer to the transient force, which dissipates energy through the good mobility of polymer chains and hydrogen bonding, resulting in a significant mitigation of the bending deformation and tearing failure of the PU-filled SCPA panels. In addition, the PU coating layer also showed a significant deformation reduction effect on the porous materials, similar to the metallic plate-based composite structures [15,16,17]. Under the action of the impact load, the polyurea molecular chain changes from the curled state to the extended state, extending in the direction of stretching, thus dissipating the energy. In addition, the peeling failure between the coating layer and the SPCA skeleton was also observed at the local deformation zone, indicating that the bonding strength should be seriously considered in strengthening the panel. However, the SEM image of the coating layer after the impact tests also showed that the micro-pores generated in the spraying process reduced the tensile strength of the layer, thus decreasing the strengthening effect of the layer on the porous materials in both deformation restraint and penetration resistance. It demonstrates that optimizing the spraying process and reducing the porosity of the coating layer are important issues to improve its strengthening performance.

In total, the anti-penetration mechanisms of SCPA and PSCPA can be illustrated by Figure 12b,c, respectively. The porous structure has low resistance to the shear stress caused by the projectile impact, thus producing a slight diffused penetration channel and shear failure band around the channel. The sheared cell fragments from the penetration channel accumulated at the front of the projectile and further caused tear–shear complex failure in cells at the back part of the panel. In consequence, the outlet diameter is slightly enlarged compared to the impact entrance. Considering that the rotation effect caused by the porous structure on the projectile was largely dependent on the penetration depth, the shear plug effect dominated the failure mode for SCPA, as illustrated in Figure 12b.

But for the PSCPA panel, the crushed cells and the PU fillers formed a compaction plug block in front of the projectile and increased the lateral damage range with the penetration depth. More cells were involved in the damage zone, dissipating the impact energy. The wearing force and the compression strength of the PU fillers are also conducive to an increase in the resistance force. The stress dispersion mechanism caused by the increased contact interface can significantly restrain the breakout of the coating layer, thus dissipating energy by generating large local tension deformations, as shown in Figure 12c. The filling of PU in the metal skeleton rapidly forms a compression environment and diffuses to a larger contact area with the coating layer. The coating layer converts the contact force into a transient stretching deformation, which forms a transition from the rubbery to the glassy state, accompanied by significant energy absorption and strength improvement. The backside support effect on the skeleton layer leads to better protection performance than the frontal coating condition.

## 5. Conclusions

In order to explore protective structures possessing high anti-penetration and energy absorption capacities in lightweight structure design for the combined action of blast waves and fragments, interpenetrating phase composites (IPCs) comprised of metallic skeleton, polyurea filler, and a coating layer were fabricated. The anti-penetration performance of the cylindrical fragment was experimentally studied based on low-velocity impact tests. The strengthening mechanism of the penetration resistance of polyurea to the SCPA plate was analyzed. The main conclusions are as follows.

(1)Polyurea with higher strength and toughness showed a better strengthening effect on the anti-penetration performance of the SCPA plate. The initial impact failure strength of the composite panel was about 2.2–3.0 times that of SCPA, and the average resistance force was improved by 108–218% and 125–274% for flat nose and spherical nose fragment penetration, respectively.(2)The backside coating method showed better anti-penetration resistance than the frontal side coating because of its backside support effect. The coating layer at the back surface is subjected to the combined actions of the localized indentation force caused by the fragment and the global bending of the polyurea-filled SCPA plate. But the layer rapidly converted the input energy from the SCPA plate into large stretching deformations, accompanied by phase transitions and energy dissipation.(3)The filling polyurea in SCPA increased the damage area of the metallic skeleton and formed a compression cone for the backside coating layer, leading to a significant stress diffusion effect. The plug block effect of the IPC and the backside support effect of the PU coating layer synergistically improved the anti-penetration performance of the plate.

## Figures and Tables

**Figure 1 polymers-16-01249-f001:**
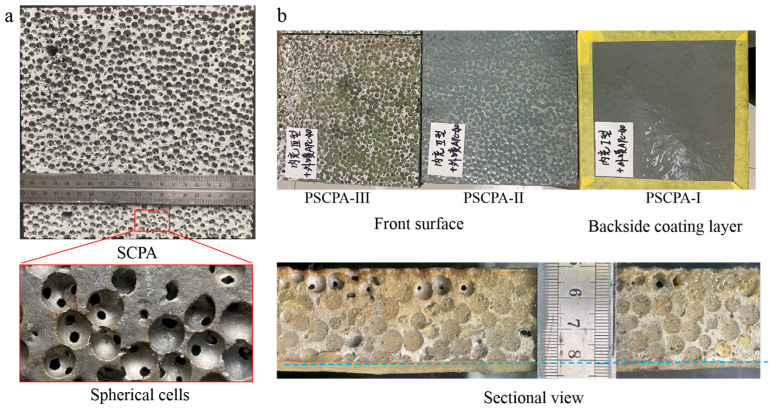
Testing samples, (**a**) spherical cell porous aluminum plate and (**b**) polyrea-filled spherical cell porous aluminum composite panel.

**Figure 2 polymers-16-01249-f002:**
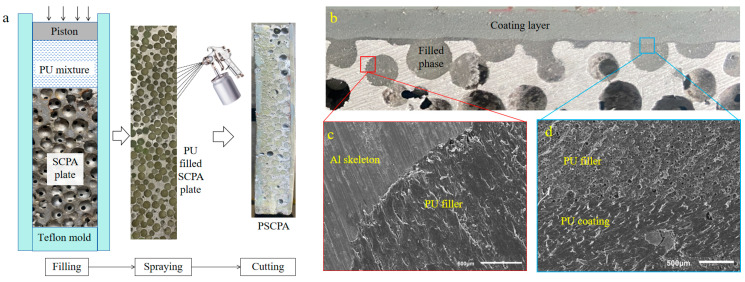
Illustration on fabrication process of PSCPA panles, (**a**) filling and spraying process, (**b**–**d**) the bonding interface between different components in PSCPA.

**Figure 3 polymers-16-01249-f003:**
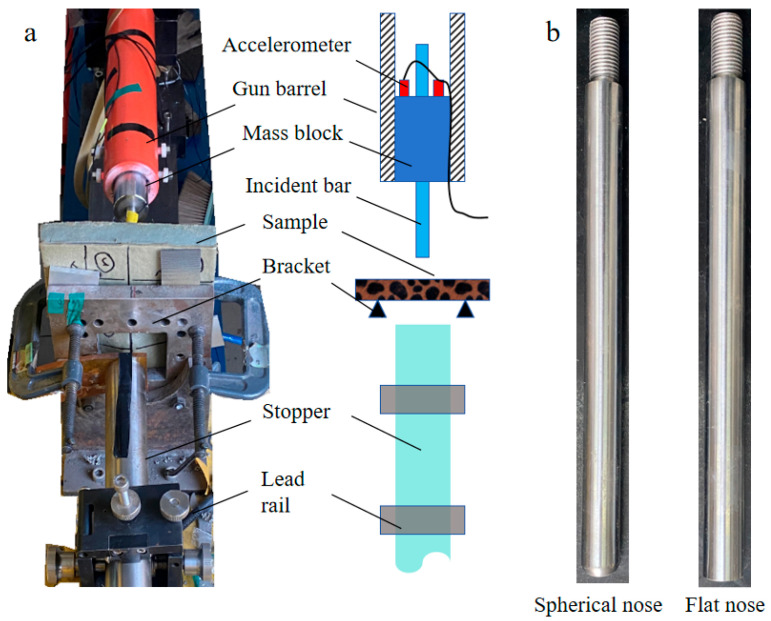
Experimental apparatus, (**a**) projectile launching system and (**b**) two different incident bars.

**Figure 4 polymers-16-01249-f004:**
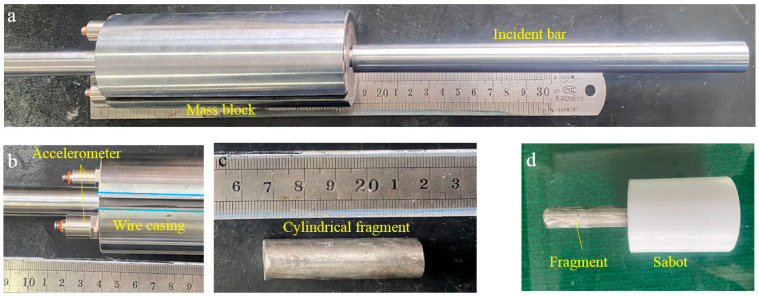
Projectile system, (**a**) assembly of mass block and the incident bar, (**b**) the accelerometer at the bottom of the mass block, (**c**) cylindrical fragment and (**d**) the assembly of fragment and the sabot.

**Figure 5 polymers-16-01249-f005:**
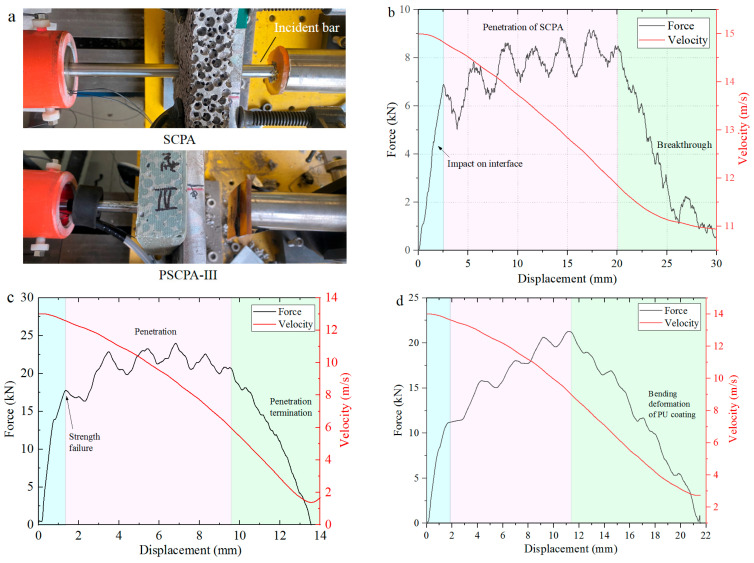
Typical experimental results, (**a**) damage of the panels, the resistant force of (**b**) SCPA and (**c**) PSCPA-III panels under flat nose incident bar impact, (**d**) the spherical nose bar to PSCPA-III panel.

**Figure 6 polymers-16-01249-f006:**
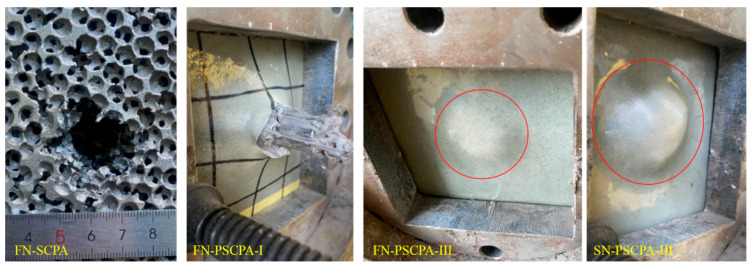
The backside view of SCPA and PSCPA panels after low-velocity penetration tests.

**Figure 7 polymers-16-01249-f007:**
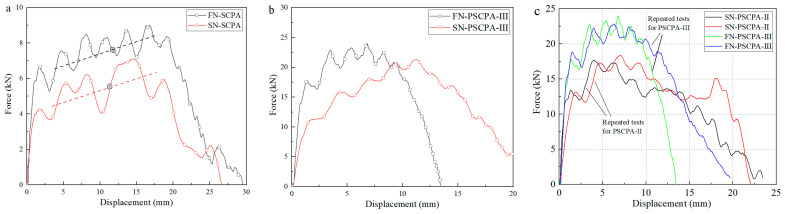
Comparison of penetration resistance between flat and spherical nose incident bar, (**a**) SCPA and (**b**) PSCPA-III, (**c**) the typical repeated test results for PSCPA panels.

**Figure 8 polymers-16-01249-f008:**
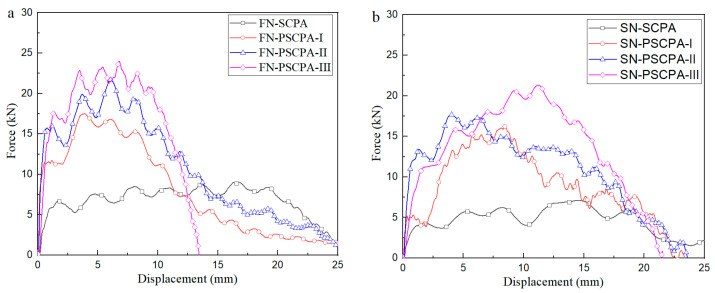
Comparison of the resistance force between different PU fillers, (**a**) flat nose penetration and (**b**) spherical nose penetration.

**Figure 9 polymers-16-01249-f009:**
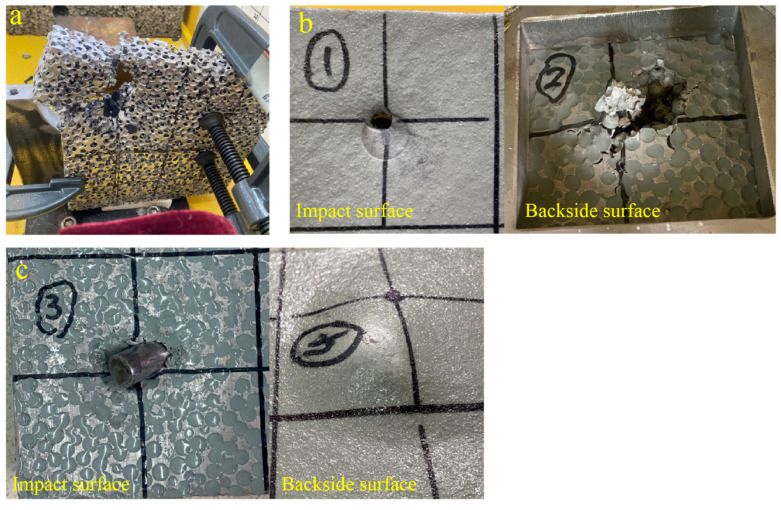
High-velocity fragment impact results. (**a**) SCPA, (**b**) PSCPA-III under reverse impact, and (**c**) PSCPA-III under forward impact.

**Figure 10 polymers-16-01249-f010:**
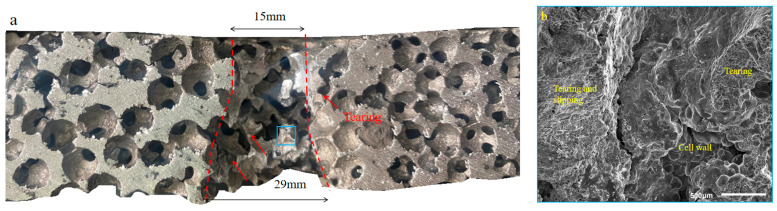
Sectional view and failure pattern of the penetration channel for SPCA plate, (**a**) penetration failure and (**b**) tearing failure mechanism of aluminum matrix in the blue box region.

**Figure 11 polymers-16-01249-f011:**
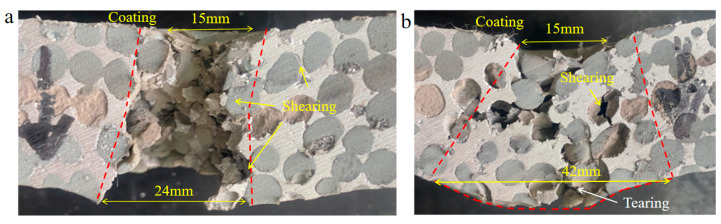
Reverse impact of fragment on PSCPA-III (**a**) *v*_0_ = 170 m/s, (**b**) *v*_0_ = 81 m/s, and (**c**) the micro-scanning image of the coating fracture surface. The red dot line indicates the failure region of the composite plate.

**Figure 12 polymers-16-01249-f012:**
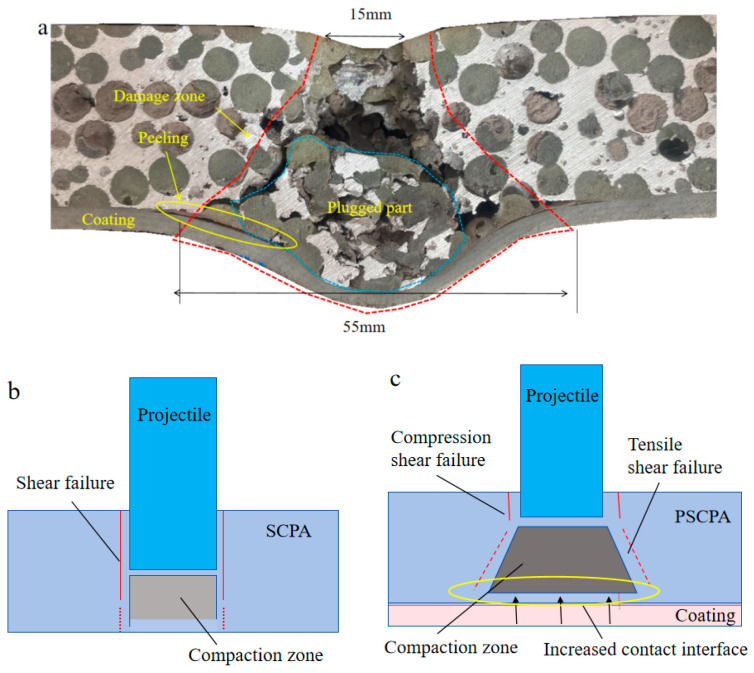
Forward impact results of PSCPA-III and analysis on anti-penetration mechanism, (**a**) failure pattern of PSCPA-III under forward impact of fragment, illustration of penetration mechanism of (**b**) SCPA and (**c**) PSCPA.

**Table 1 polymers-16-01249-t001:** Parameters of three polyurea materials.

Polyurea	Solid Content	Density (kg/m^3^)	Tensile Strength (MPa)	Tear Strength (kN/m)	Elongation
APC-20	≥90	1050	15	85	≥350
APC-30	≥96	1050	22	100	≥300
APC-40	≥96	1070	32	110	≥300

**Table 2 polymers-16-01249-t002:** Summary of the average resistance force for different specimens.

	SCPA	PSCPA-I	PSCPA-II	PSCPA-III
FN	7.14 ± 0.6	14.83 ± 1.3	19.77 ± 1.6	22.68 ± 1.7
SN	5.48 ± 0.7	12.36 ± 1.5	15.41 ± 1.8	20.05 ± 1.4

## Data Availability

Data are contained within the article.

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
