# Peer review of "Strengthening Mechanism of Polyurea to Anti-Penetration Performance of Spherical Cell Porous Aluminum"

_polymers, 2024, doi:10.3390/polym16091249_

Round 1
Reviewer 1 Report
Comments and Suggestions for Authors
The manuscript presents penetration tests of a metallic skeletal IPC with polyurea filling and coatings under low velocity impact of large mass fragments. Overall, the manuscript it is well structured and well written, however I have some comments to be addressed for revision:
1. What it is designed for this material? In the manuscript it is not clearly explained the purpose of the tested material.
2. For the evaluation of the penetration capacity, how many times were the tests repeated under the same conditions? Can you provide some additional information regarding this, just to evaluate how representative are the results?
Author Response
1. What it is designed for this material? In the manuscript it is not clearly explained the purpose of the tested material.
Response:The filling and coating of polyurea elastomer for porous aluminum panel can provide a new approach to improve the anti-penetration performance of energy absorption layers in protective structures. The composite plate containing both metallic skeleton and polyurea coatings could provide both anti-blast and anti-penetration capacities, showing high potential in armor design for personal protective equipment and light armored vehicles. We have demonstrated the purpose and objectives of this work in the Introduction section of the revised manuscript according to your comments.
2. For the evaluation of the penetration capacity, how many times were the tests repeated under the same conditions? Can you provide some additional information regarding this, just to evaluate how representative are the results?
Response:Two repeated experiments were performed for each testing conditions to acquire the average resistant force and accurately evaluate the anti-penetration capacity of different panels. The repeated testing results was supplied in Section 3.1. The average resistant force and the testing difference was summarized in Table 2 while typical curves were analyzed in the following section.
Reviewer 2 Report
Comments and Suggestions for Authors
The behavior of composite structures under various influences is a current area of research. Compositions of various types are used in most fields of science. The question of the compositional structures reaction to external influences is interesting. The resistance of cellular structures filled and coated with polyurea to penetration of large mass fragments is considered in the work. The question of strengthening the protective characteristics of research objects is being considered. The work reflects a number of important results. It is well structured.
Notes and recommendations:
1. The “Research objectives” subsection needs to be added to the “Introduction” to improve understanding and ease of reading of the article.
2. I would like to see more detailed information about the scope of application of the research objects under consideration in the “Introduction” section.
3. The section “Materials and Methods” requires improvement. The experimental design and methodology need to be described in more detail. I would like to see information on how many times the experiment was repeated for different samples. Information about the experimental equipment must be reflected: brand, manufacturer, main performance characteristics.
4. There is no comparison with the results of other authors in the “Discussion” section. I would like to see a critical assessment of the results. A section related to the main limitations of the presented studies is worth adding to the discussion. Further directions for the scientific research development need to be reflected.
Author Response
1. The “Research objectives” subsection needs to be added to the “Introduction” to improve understanding and ease of reading of the article.
Response: The filling and coating of polyurea elastomer for porous aluminum panel can provide a new approach to improve the anti-penetration performance of energy absorption layers in protective structures. The composite plate containing both metallic skeleton and polyurea coatings could provide both anti-blast and anti-penetration capacities, showing high potential in armor design for personal protective equipment and light armored vehicles. We have added the research objectives in the Introduction section of the revised manuscript, as marked in red part.
2. I would like to see more detailed information about the scope of application of the research objects under consideration in the “Introduction” section.
Response: The experimental results could help to develop advanced anti-strike panels possessing high protection performance from blast loading and fragment penetration, which is of high potential in structure design of the personal protective equipment and lightweight armored vehicles. We have added the demonstration in the Introduction section of the revised manuscript according to your comments.
3. The section “Materials and Methods” requires improvement. The experimental design and methodology need to be described in more detail. I would like to see information on how many times the experiment was repeated for different samples. Information about the experimental equipment must be reflected: brand, manufacturer, main performance characteristics.
Response:Two repeated experiments were performed for each testing conditions to acquire the average resistant force and accurately evaluate the anti-penetration capacity of different panels. The repeated testing results was supplied in Section 3.1. The average resistant force and the testing difference was summarized in Table 2 while typical curves were analyzed in the following section. In addition the detail information of the materials and experimental equipment was supplied in this section according to your comments.
4. There is no comparison with the results of other authors in the “Discussion” section. I would like to see a critical assessment of the results. A section related to the main limitations of the presented studies is worth adding to the discussion. Further directions for the scientific research development need to be reflected.
Response:Thank you for your comments and suggestions. We have added the comparison with the current related studies in the discussion section. However, according to the literature review, we found that study on strength mechanism of polyurea coating to anti-penetration performance of porous materials is very limited. Large amount of related studies are limited to metallic plates, ceramic panels and thin-walled studies. Direct comparison of the anti-penetration performance is difficult because of the base materials difference. Therefore we compared the strengthen mechanism between different matrix materials. Also we added the limitations and future study in the discussion section according to your comments. Than you again!
Reviewer 3 Report
Comments and Suggestions for Authors
1. What is the identified literature gap? How this work differs from the existing literature?
2. In Table 1 check the unit kg/m3 which should be kg/m^3.
3. In section 2.1 provide the schematic illustration of fabricating the SCPA.
4. It is given that, “the thickness 140 of the coating layer was 3.0mm”. How the thickness was measured and controlled?
5. Provide the photographic view of spraying polyurea on SCPA.
6. State the potential applications of this study.
7. Provide SEM images before and after coating and also the SEM taken at the impacted surface.
Author Response
1.What is the identified literature gap? How this work differs from the existing literature?
Response:According to literature review, we found that composite structure containing both energy absorption layers and anti-penetration layers have attracted great interests due to its excellent protective capacity to blast and impact conditions. Porous materials usually have high energy absorption performance but the anti-penetration capacity to fragments and projectiles is relatively weak due to its porous structure and low density. Analysis on current studies shown that research on the strengthen techniques of anti-penetration performance for porous materials is very limited. Besides it was shown that studies on strengthen mechanism of PU coatings were largely limited with solid materials, such as metallic plates, ceramic panels and thin-walled structures. The enhancement mechanism of polyurea coating to energy absorption materials and the preparation method to improve the bonding interface strength need to be studied urgently to develop advance protective structures. We used polyurea coating and filling phase to strengthen the penetration resistance of SCPA to large mass fragments. We proposed an approach to increase the bonding strength between PU coating layer and the porous materials, thus improving the strengthen effect of the coating layer. Also we identified the failure patterns and strengthen mechanisms of composite panels with different configurations. We have added this analysis to the revised manuscript according to your comments.
2.In Table 1 check the unit kg/m3 which should be kg/m^3.
Response: We have corrected the unit in Table 1.
3. In section 2.1 provide the schematic illustration of fabricating the SCPA.
Response: The fabricating process of SCPA and PSCPA panels is shown in Fig. 2 in the revised manuscript.
4. It is given that, “the thickness of the coating layer was 3.0mm”. How the thickness was measured and controlled?
Response: The thickness increasing velocity could be controlled by the spraying rate and the spraying times The coating layer was formed by multiple spraying process until the thickness reached 3mm. We have added the demonstration in the revised manuscript.
5. Provide the photographic view of spraying polyurea on SCPA.
Response: The fabricating process of SCPA and PSCPA panels is shown in Fig. 2 in the revised manuscript.
6. State the potential applications of this study.
Response: The experimental results could help to develop advanced anti-strike panels possessing high protection performance from blast loading and fragment penetration, which is of high potential in structure design of the personal protective equipment and lightweight armored vehicles. We have added the demonstration in the Introduction section of the revised manuscript according to your comments.
7. Provide SEM images before and after coating and also the SEM taken at the impacted surface.
Response: According to your comments, SEM images of aluminum matrix, filling PU and the coating layers before and after the impact tests are supplied, as seen in Figures 2, 10 and 11.
Reviewer 4 Report
Comments and Suggestions for Authors
Dear Authors,
Your porous components are completely missing an examination of the theoretical and experimental porosity values. Include CT scan analysis for correct measurement of your experimental components, and your research will be valuable.
In general, porous components should be evaluated for flammability, water absorption, and sound absorption to determine their effective strength.
Author Response
1. Your porous components are completely missing an examination of the theoretical and experimental porosity values. Include CT scan analysis for correct measurement of your experimental components, and your research will be valuable.
Response: The analysis on porosity of SCPA is added in Section 2.1, including the theoretical and experimental values. Thank your for your comments.
2. In general, porous components should be evaluated for flammability, water absorption, and sound absorption to determine their effective strength.
Response: Thanks for your comments. We certainly believe that more tests on the flammability, water and sound absorption performance could largely promote its application potential in engineering protection. However, as demonstrated in introduction section, the main purpose of this work was to evaluate the anti-penetration performance of polyurea coated porous aluminum panels to large mass fragments. Thus the penetration tests were mainly considered, the anti-penetration mechanism the the composite panel and the strengthening mechanism of polyurea filler and coating layers were analyzed. At last, the strengthening mechanism including the phase transition effect of the backside coating and the plug block effect was summarized to provide more suggestions on the design of the protective structure containing both energy absorption layer and anti-penetration layers. According to your comments, more tests including the heat resistance, water and sound absorption will be performed on the composite panels to improve its comprehensive performance in our future work.